# Structure and inhibition mechanisms of *Mycobacterium tuberculosis* essential transporter efflux protein A

Nitesh Kumar Khandelwal [1,7,8], Meghna Gupta [1,7,8], James E. Gomez[2], Sulyman Barkho[2], Ziqiang Guan [3], Ashley Y. Eng[2], Tomohiko Kawate[2], Sree Ganesh Balasubramani[4], Andrej Sali [4], Deborah T. Hung [2,5,6] ✉ & Robert M. Stroud [1] ✉

A broad chemical genetic screen in *Mycobacterium tuberculosis* (*Mtb*) identified compounds (BRD-8000.3 and BRD-9327) that inhibit the essential efflux pump EfpA. To understand the mechanisms of inhibition, we determined the structures of EfpA with these inhibitors bound at 2.7-3.4 Å resolution. Our structures reveal different mechanisms of inhibition by the two inhibitors. BRD-8000.3 binds in a tunnel contacting the lipid bilayer and extending toward the central cavity to displace the fatty acid chain of a lipid molecule bound in the apo structure, suggesting its blocking of an access route for a natural lipidic substrate. Meanwhile, BRD-9327 binds in the outer vestibule without complete blockade of the substrate path to the outside, suggesting its possible inhibition of the movement necessary for alternate access of the transporter. Our results show EfpA as a potential lipid transporter, explain the basis of the synergy of these inhibitors and their potential for combination anti-tuberculosis therapy.

*Mycobacterium tuberculosis (Mtb)*, the causative agent of human tuberculosis, remains a major global health threat, responsible for 1.3 million deaths in 2022[1]. Mortality rates from tuberculosis are the highest for any bacterial disease[1]. New antibiotics are needed to combat *Mtb* since drug resistance constantly erodes the efficacy of currently used antibiotics. Several first-line antibiotics used against *Mtb*, including isoniazid and ethambutol affect the cell wall, which provides the first line of defense against the environment within alveolar macrophages. Seeking innovative approaches to target novel vulnerabilities in *Mtb*, we recently developed a broad chemical genetic screening strategy termed PROSPECT (**pr**imary screening **o**f **s**trains to **p**rioritize **e**xpanded **c**hemistry and **t**argets), in which compounds were

screened against pools of strains each depleted of a different essential target. Using this approach for generating chemical genetic interaction profiles between small molecules and the set of genetic hypomorphic mutants, we identified chemical scaffolds that kill *Mtb* by inhibiting unconventional protein targets including the essential efflux pump A (EfpA)[2]. This strategy enabled our initial identification of BRD-8000 as an inhibitor of EfpA. We performed chemical optimization to achieve potent whole-cell activity against wild-type *Mtb* for the analogues BRD-8000.1, BRD-8000.2 and BRD-8000.3, with MIC$_{90}$ of 12.5 μM, 3 μM and 800 nM, respectively (Fig. 1a). Resistance to BRD-8000.3 could be conferred by mutations in EfpA, including substitutions V319F and A415V. BRD-8000.3 is an uncompetitive inhibitor of EfpA efflux of

[1]Department of Biochemistry and Biophysics, University of California San Francisco, San Francisco, CA, USA. [2]Broad Institute of MIT and Harvard, Cambridge, MA, USA. [3]Department of Biochemistry, Duke University Medical Center, Durham, NC, USA. [4]Department of Bioengineering and Therapeutic Sciences, University of California San Francisco, San Francisco, CA, USA. [5]Department of Genetics, Harvard Medical School, Boston, MA, USA. [6]Department of Molecular Biology and Center for Computational and Integrative Biology, Massachusetts General Hospital, Boston, MA, USA. [7]Present address: Department of Chemical Physiology and Biochemistry, Oregon Health & Science University, Portland, OR, USA. [8]These authors contributed equally: Nitesh Kumar Khandelwal, Meghna Gupta. ✉e-mail: hung@molbio.mgh.harvard.edu; stroud@msg.ucsf.edu

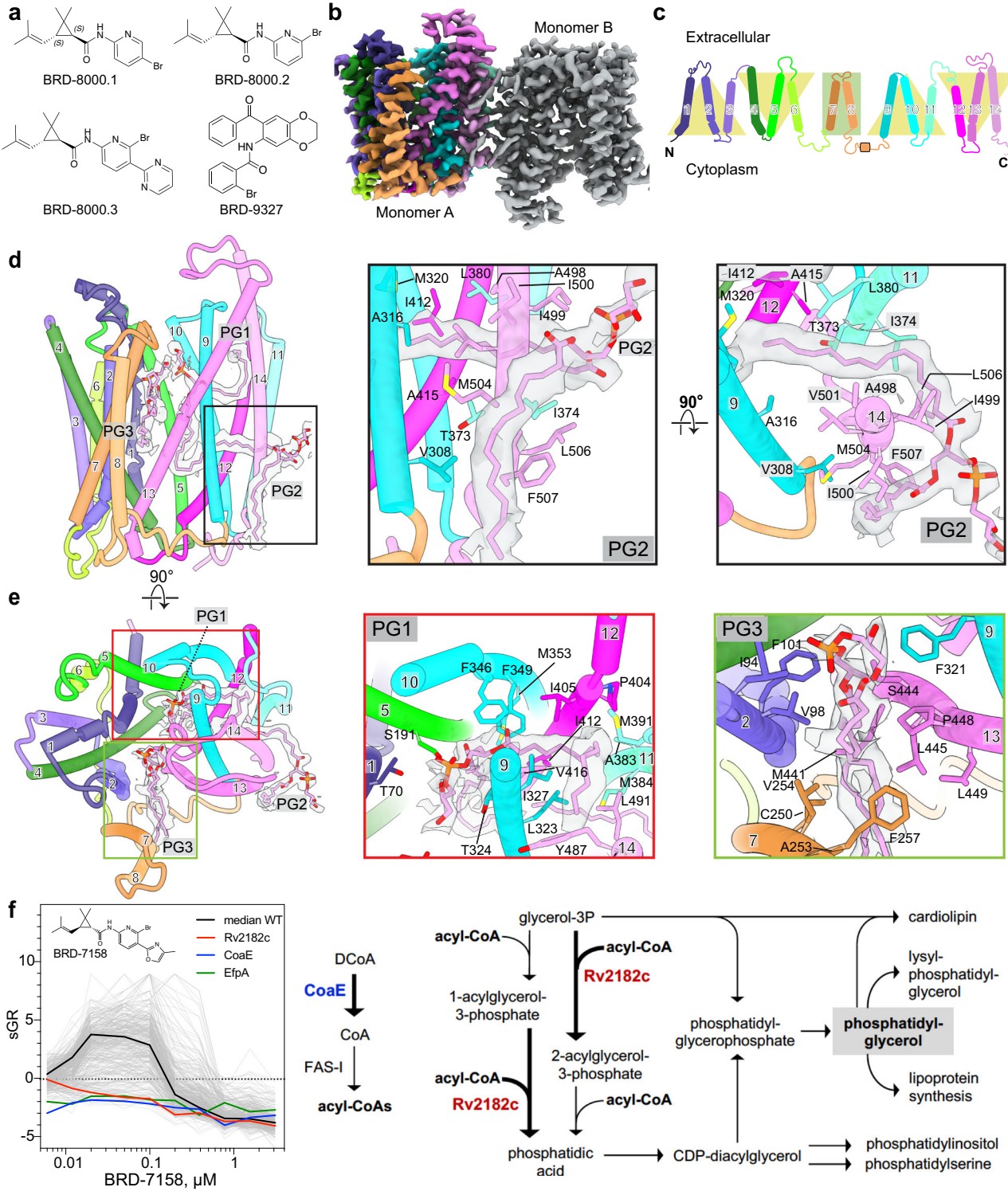

ethidium bromide (EtBr), a demonstrated substrate, although the natural substrate and function of EfpA is not yet identified. Leveraging the same PROSPECT strategy, we subsequently identified a structurally distinct small molecule EfpA inhibitor BRD-9327[3] (Fig. 1a), based on it sharing a similar chemical-genetic interaction profile with BRD-8000.3. Biochemical evidence revealed that BRD-8000.3 and BRD-9327 work synergistically and display collateral sensitivity with distinct mechanisms of resistance[3].

EfpA is an essential transporter in *Mtb* and categorized as a member of the QacA (quaternary ammonium compound A) transporter family[4]. Although EfpA has been suggested to play a role in *Mtb*

drug resistance[5], its essentiality in axenic culture implies that antibiotic efflux is a secondary role, and that the primary function of EfpA is the transport of a yet-uncharacterized biomolecule across the *Mtb* plasma membrane. A better understanding of how the small molecules BRD-8000.3 and BRD-9327 interact with EfpA to prevent transport will support both the potential development of related molecules and the discovery of new classes of inhibitors. Hence, we sought to determine the mechanisms of action of these inhibitors on EfpA by determining structures of EfpA in complex with these compounds.

In this study we determined the mechanism of inhibition of the EfpA transporter by two different inhibitors. To facilitate image

**Fig. 1 | Structure of EfpA. a** Chemical structures of the BRD-8000 series compounds and BRD-9327 identified by PROSPECT. **b** The cryo-EM density map for the antiparallel dimer of EfpA$^{EM}$. **c** Schematic representation of the topological arrangement of EfpA. Three transmembrane helix (TM) bundles are related to each other by a twofold pseudosymmetry, with two extra linker helix (TM7 and 8). **d** EfpA monomer A atomic model along with PG molecules (PG1, PG2 and PG3) with corresponding EM density (left panel). PG2 molecule interaction (middle panel) and rotated 90° forward from the position in the middle panel (right panel). The acyl chain of PG2 that is outside of the protein runs parallel to the TM14 against hydrophobic residues (F507$^{TM14}$, L506$^{TM14}$, M504$^{TM14}$ and I500$^{TM14}$) and reaches downward to the cytoplasmic surface. The head group present between the dimer interface and its second acyl chain enters EfpA via space present between the G377$^{TM11}$, G502$^{TM14}$ residues. This acyl chain is perpendicular to the TM9, TM11, TM12 and TM14 and interacts with M320$^{TM9}$, A316$^{TM9}$, T373$^{TM11}$, I374$^{TM11}$, L380$^{TM11}$, V501$^{TM14}$, A498$^{TM14}$, I499$^{TM14}$, I412$^{TM12}$, A415$^{TM12}$, V416$^{TM12}$ and L419$^{TM12}$. **e** EfpA monomer (left hand panel **d**) rotated 90° forward (left panel). PG1 molecule (middle panel) lies between TM9 (I313$^{TM9}$, A316$^{TM9}$, M320$^{TM9}$, L323$^{TM9}$, I327$^{TM9}$ and Y330$^{TM9}$), TM14 (Y487$^{TM14}$ and L491$^{TM14}$), TM11 (L380$^{TM11}$, M384$^{TM11}$ and A383$^{TM11}$), TM12 (I405$^{TM12}$, P404$^{TM12}$, I412$^{TM12}$ and V416$^{TM12}$), TM10 (F346$^{TM10}$, F349$^{TM10}$ and M353$^{TM10}$) TM5 (S191$^{TM5}$) and TM1 (T70$^{TM1}$). PG3 (right panel) lies between TM2 (I94$^{TM2}$, V98$^{TM2}$ and F101$^{TM2}$), TM7 (C250$^{TM7}$, A253$^{TM7}$, V254$^{TM7}$ and F257$^{TM7}$), TM9 (F321$^{TM9}$) and TM13 (M441$^{TM13}$, S444$^{TM13}$, L445$^{TM13}$, P448$^{TM13}$ and L449$^{TM13}$). **f** Profiles of 361 strains in pooled PROSPECT against BRD-8000 analogue BRD-7158; each strain is shown as a grey line with relevant sensitized strains labeled as in the legend- WT (black), CoaE (blue), Rv2812c (red) and EfpA (green). The x-axis is concentration of BRD-7158 in μM, the y-axis is standardized growth rate (sGR) of each strain across 8 doses tested. The right panel shows the pathway of PG synthesis highlighting the functions of CoaE and Rv2812c.

processing EfpA was augmented by addition of a BRIL domain at the N-terminal end followed by optimized anti-BRIL Fab, and a nanobody against the Fab. Structures were determined by using cryo-electron microscopy (cryo-EM). EfpA was maintained in detergent micelles and assembled as dimers arranged in antiparallel (2.7 Å resolution) and parallel (3.2 Å resolution) configuration. The structures show lipid molecules within channels inside EfpA. This suggests that its yet undetermined function might be as a lipid transporter. To further elaborate this functional role 361 mutant strains of *Mtb* were tested for analogue inhibitor BRD-7158 that sensitize EfpA and other PG synthesis pathway genes. The interactions support the concept that EfpA may act as a transporter of phosphatidyl glycerol (PG) from inner to outer leaflet of the plasma membrane. To understand the inhibition mechanisms, we determined cryo-EM structures of EfpA inhibited by two inhibitors BRD-8000.3 (3.45 Å resolution), and BRD-9327 (3.0 Å resolution). BRD-8000.3 binds perpendicular to the transmembrane domains (TM) and blocks a tunnel that previously bound the fatty acid chain of a phospholipid. BRD-9327 occupies the opening towards the extracellular cavity of the protein to block the transport function. Finally, to understand the synergistic action of these inhibitors, we determined structure of EfpA with both BRD-8000.3 and BRD-9327 exhibiting binding of each at two different locations simultaneously in each protomer. These structures identify the possible function of EfpA and elucidate different modes of inhibition by these two inhibitors and support the idea that molecules that inhibit the same target could be used synergistically in combination to treat tuberculosis.

## Results

### Structure determination of EfpA

*Mtb* EfpA is a 55.58 kDa protein with 14 transmembrane domains (TM). We codon optimized EfpA from 3 species of mycobacteria: *Mtb*, *Mycobacterium marinum* and *Mycobacterium smegmatis* for expression screening in *Escherichia coli* (details in Methods). Most of the structure is contained in the TM regions with minimal rigid extracellular domains, therefore making it a challenging target for image alignment and structural determination by cryo-EM. To facilitate alignment, we replaced 48 amino acids that preceded the N-terminal TM1 domain and were predicted to be disordered by Alphafold2[6], by a 12 kDa four-helix soluble domain of BRIL, a variant of apocytochrome b$_{562}$ engineered for stability[7] (Supplementary Fig. 1a). Fab fragments have been optimized to bind to BRIL providing further prospects for correctly classifying images for three-dimensional reconstruction[8]. Nanobodies that bind to these pre-optimized anti-BRIL Fab fragments have also been optimized and made available, thus providing a common strategy of pre-optimized components that make for efficient structure determination from proteins whose size or structural orientation make them difficult to align[9]. Based on predicted models from Alphafold2 we also introduced a mutation P171R in the loop between TM4 and TM5 of EfpA with the intention that it might act as a site of salt bridge formation with nearby acidic residues (D24, E26, E30,

D72 and E71) of the BRIL molecule. The resulting engineered EfpA$^{EM}$ has similar activity to wild-type EfpA as measured by comparison of transport of EtBr, with that of wildtype EfpA (Supplementary Fig. 1b). The EfpA$^{EM}$ construct was used for protein expression (Supplementary Fig. 1c), and a tripartite complex consisting of purified protein, anti-BRIL Fab and anti-Fab nanobody (Nb) was prepared for cryo-EM[9] (Supplementary Fig. 1d).

EfpA$^{EM}$ purified as a dimeric assembly, and in single particle analysis of cryo-EM data the dominant arrangement of monomers in the dimer was anti-parallel seen at 2.7 Å resolution (Supplementary Figs. 2 and 3), along with less frequent parallel dimer association seen at 3.2 Å resolution (Supplementary Fig. 2 and 4). The dimeric assembly may be a result of rearrangement during purification and concentration in detergent lipid micelles or may reflect a possible physiological association. This phenomenon has also been observed in other major facilitator superfamily (MFS) transporters[10,11]. We determined the structures of the antiparallel dimer EfpA$^{EM}$ to 2.7 Å resolution and the parallel dimer to 3.2 Å resolution (Supplementary Figs. 2–4). Complexes were formed with inhibitors BRD-8000.3, and with BRD-9327 developed from the chemical genetic screen[2,3]. Their structures were determined in the antiparallel dimer configuration to 3.45 Å and 3.0 Å resolution, respectively (Figs. 2 and 3).

### Overall structure

All four determined structures of EfpA$^{EM}$ are in an outward open conformation with an extracellular gate between TM1, 2, 5 and TM9, 10 open while the intracellular gate was closed by regions of TM4 and TM12, 13 (Fig. 1b). The structure is well ordered for all 14 TM domains and loops (residues 49-518) with exception of 12 disordered amino acids at the C-terminus (Fig. 1b–d). EfpA TM helices organize in a typical MFS transporter fold where two bundles formed by 6 helices (TM1-TM6 and TM9-TM14) are arranged with an internal twofold pseudo symmetry (Fig. 1c). TM7 and TM8 form a linker in the long loop between the two bundles and are juxtaposed next to the TM2 of EfpA (Fig. 1d, e).

In the antiparallel configurations of the dimer seen at 2.7 Å resolution, amino acids from TM11 and TM14 of both monomers form the dimer interface. A 'prime' after the sequence number indicates amino acids from the second monomer. Y378-Y378′ form p−pi interactions and I499-I499′ form hydrophobic contacts across the twofold axis between molecules. These are bolstered by interactions L385-L506′-I374′ above, and L385′-L506-I374 below the twofold axis on these two helices to form the dimerization interface between molecules (Supplementary Fig. 5a). The structures of the antiparallel dimer are therefore twofold symmetric assemblies in which the twofold axis lies in the plane of the bilayer with the same side chains from each molecule displayed in antiparallel arrangement and interdigitated in the interface between molecules above and below the twofold axis. The parallel dimer uses the same interstitial residues that interdigitate between identical residues in the sequence in a side-by-side manner,

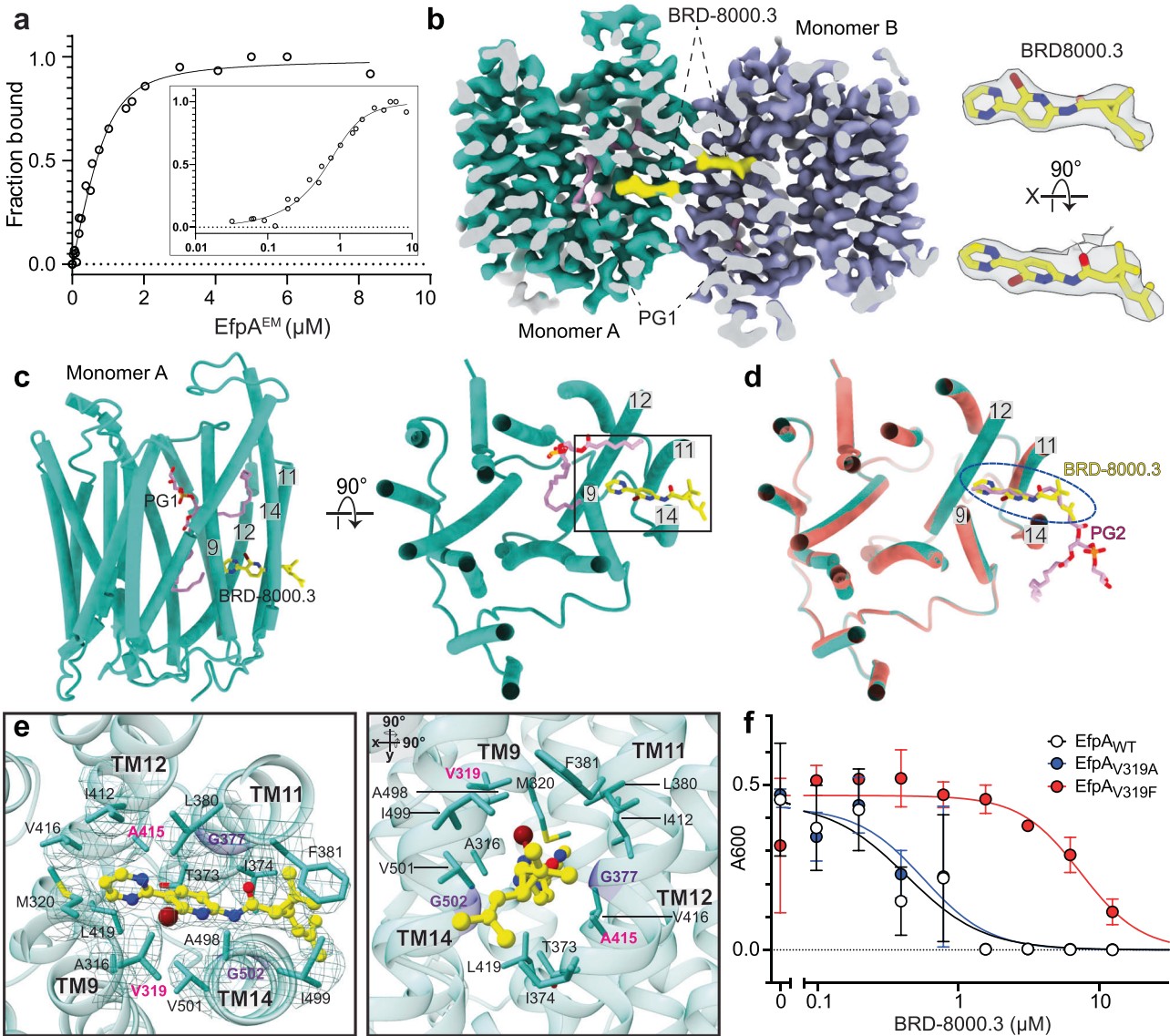

**Fig. 2 | Mode of inhibition of EfpA by BRD-8000.3. a** BRD-8000.3 binds to EfpA$^{EM}$ in a ligand-observed NMR assay. Intensity of the inhibitor's aromatic protons is monitored in a titration series of EfpA$^{EM}$ (x-axis), in which peak heights are converted to fractional occupancy (y-axis). A dissociation constant is determined by fitting the data to a standard bimolecular binding model (see section "Methods"). Data represents 3 independent titrations. **b** Section of cryo-EM map of BRD-8000.3 (yellow) bound EfpA$^{EM}$ dimer (left panel). Experimentally determined cryo-EM density of BRD-8000.3 in monomer A (right panel). Map contour level = 0.597 in ChimeraX. **c** Tertiary structure of the EfpA$^{EM}$ monomer A with BRD-8000.3 (yellow) with lipid shown in pink, side view left and top view from the extracellular side of monomer A. **d** Superposition of BRD-8000.3 (yellow) bound EfpA$^{EM}$ monomer (turquoise) onto the apo structure (orange) with PG2 in apo (pink). BRD-8000.3

displaces the PG2 molecule present in the apo structure in the BRD-8000.3 bound structure. **e** Tertiary structure for BRD-8000.3 binding pocket showing BRD-8000.3 (yellow) and associated side chains viewed from the extracellular face along with cryo-EM map density as mesh (left) and side view obtained by rotating left panel at 90° on x and y axis. Entry portal glycines G377$^{TM11}$ and G502$^{TM14}$ are colored violet and side chain atoms colored by atom, nitrogen (blue), oxygen (red), bromide (brown). Residues where mutation led to resistance to BRD-8000.3 are labelled in pink. **f** Dose–response curves showing growth of *M. bovis* BCG transformed with plasmids overexpressing WT (V319), V319F, and V319A alleles of *Mtb* EfpA in response to varied concentrations of BRD-8000.3. x-axis is inhibitor concentration (μM) and y-axis is culture density after 10 days of growth (OD$_{600nm}$). Graph is plotted as mean values ± SD of 3 replicates.

resulting in a twofold symmetric assembly in which the symmetry axis is perpendicular to the membrane plane (Supplementary Fig. 6a–c).

## The structural basis for lipid recognition

We observed extra density in the cryo-EM structures of within EfpA$^{EM}$ purified from *E. coli*. Six well-defined densities line tunnels (numbered 1–6) within each monomer are consistent with aliphatic chains from three diacyl phospholipids. The characteristics of the tunnels within EfpA were also investigated using the MOLEonline server[12] applied to EfpA.

To identify any bound species, we extracted contents of the micelles and subjected them to lipidomic analysis using normal phase liquid chromatography/mass spectrometry (LC/MS). Lipid peaks were identified as corresponding to phosphatidyl glycerol (PG), cardiolipin (CDL), with much less prominent peaks identified as phosphatidyl ethanolamine (PE) as well as some free fatty acids from the *E. coli* membrane, with cholesterol hemisuccinate (CHS) and lauryl maltose neopentyl glycol (LMNG) from the micelle preparative procedures (Supplementary Figs. 5b and 7). As controls, representative negative ion MS/MS spectra of PG (16:0/17:1), PE (16:0/17:1) and CL (68:2) are

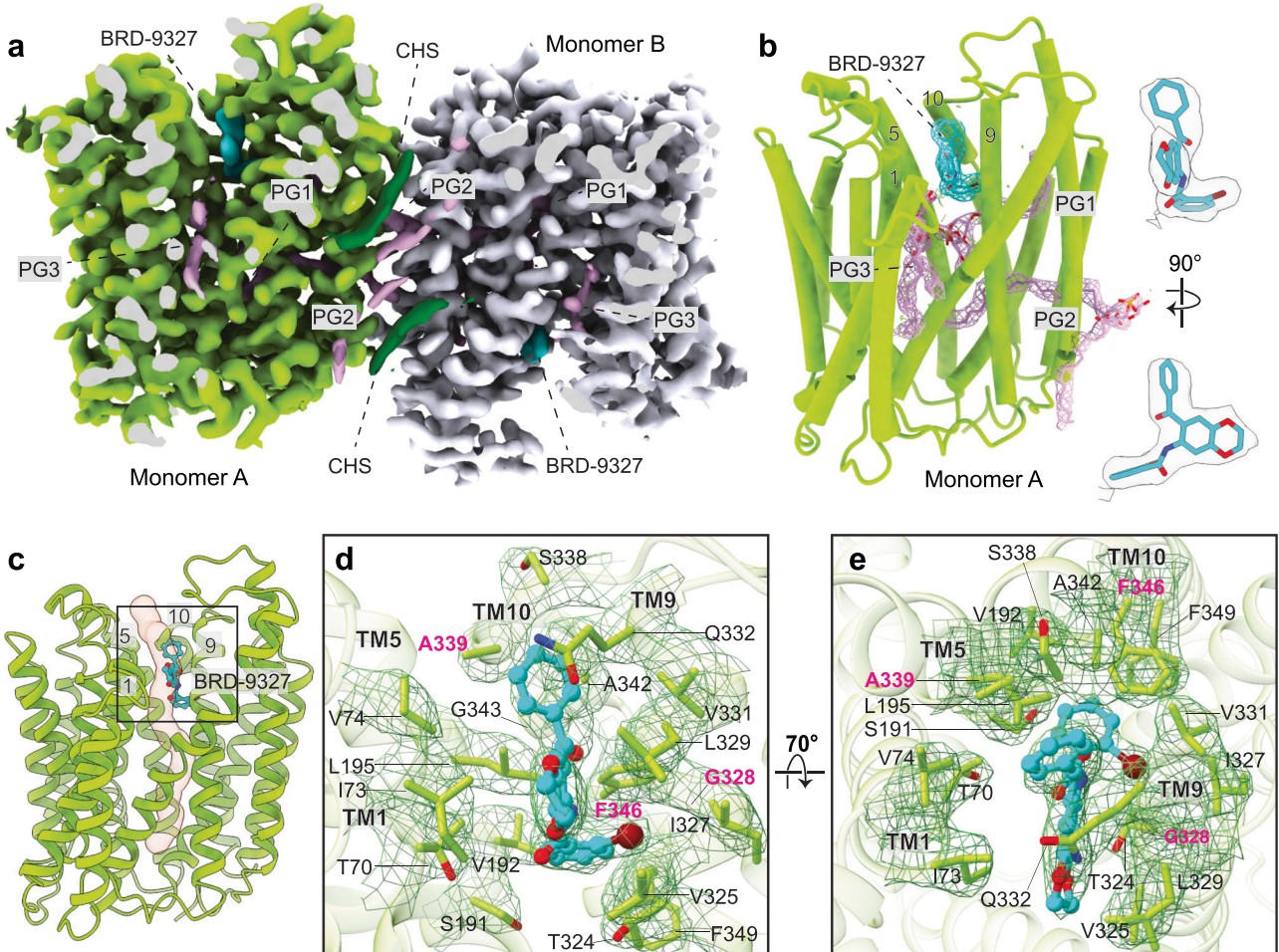

**Fig. 3 | Mode of inhibition of EfpA by BRD-9327. a** Section of the cryo-EM density map of BRD-9327 bound EfpA^EM. Both monomers of EfpA are shown as yellow-green and gray color respectively. Each monomer has one BRD-9327 molecule (aqua color) and three PG molecules (pink color). **b** Tertiary structure of EfpA^EM monomer A with BRD-9327 (aqua) bound. Density for lipids PG1, PG2 and PG3 are shown in pink (left panel). Cryo-EM density of BRD-9327 (right panel). Map contour level = 0.679 in ChimeraX. **c** Structure of BRD-9327 shows it bound in the external vestibule of EfpA^EM between TM1,5,9 and 10. **d** Model for BRD-9327 binding site in EfpA^EM oriented as in c along with cryo-EM map density as mesh. **e** Side view rotated by 70° forward from the position in the d. Residues where mutation led to resistance to BRD-9327 are labelled in pink. Color by atom Nitrogen (blue), oxygen (red) and bromide (brown).

shown (Supplementary Fig. 7). The acyl compositions are revealed by fatty acid anions shown in the product ion mass spectra.

The resolution of the lipid head group densities was not clear enough to identify them unambiguously. Mass spectrometry of the lipids uniquely determines lipids in the sample but is not quantitative as to relative amounts of phospholipids. However, in view of the relatively high abundance of the PG mass spectrometry peak versus PE, we tentatively interpreted them as three molecules of PG (therefore termed sites PG1, PG2 and PG3) which fit these densities well (Fig. 1d, e, Supplementary Figs. 5c and 8). The difference between PG and PE is two heavy atoms in the head groups in PG and would be difficult to ascertain unambiguously from cryo-EM maps at these resolutions. In theory, the distance between PG1 and PG3 is such that they could be bridged by glycerol to correspond to one, four-chain CDL.

In both the parallel and antiparallel dimers resolved by cryo-EM, the density of the two fatty acids chains of PG1 are similar and lie within the protein domain of EfpA^EM (Supplementary Figs. 6D and 8A). The phosphatidyl-head group lies in the outer vestibule. One chain lines the wall of the vestibule towards the inner leaflet while the second chain reaches between TM9, TM12, TM13, TM14 toward the outer leaflet. Both fatty acid chains are almost exclusively in contact with hydrophobic side chains from the TMs. The head group is in contact with solvent in the outer vestibule

where several polar side chains provide a hydrophilic environment (Fig. 1e).

PG2 densities differed between the parallel and antiparallel dimers. In the antiparallel dimer, one of the fatty acid chains of PG2 penetrates through the side of the protein between TM11 and TM14 away from the external head group towards the central channel. The second fatty acid chain extends from the head group outside the protein downward to the cytoplasmic surface in the region that would be occupied by the inner leaflet. The head group lies outside of the protein that would place it in contact with the inner leaflet (Fig. 1d). The head groups of PG2 from each of the two EfpA^EM monomers are in contact with each other supporting some degree of hydrophilic contact around the twofold symmetry axis between them. It is possible that a single CDL (equivalent to two PG2 bridged by glycerol) could fulfill these densities in the micellar complex though this arrangement and the antiparallel dimer itself seems unlikely to be physiological unless EfpA itself forms physiological dimers. Two molecules of CHS from the solubilization solutions fill the groove between EfpA^EM molecules on the front side (Supplementary Fig. 5c). In contrast, in the parallel dimer, the two fatty acid chains of two individual PG2 molecules of the antiparallel dimer are replaced by a single phospholipid molecule. One acyl chain is inserted in through the sides of each monomer leaving the head group in between monomers (Supplementary Fig. 6a-e). There is extra lipid density (PG4) in the

parallel dimer which lies outside the proteins with fatty acid chains reaching toward the cytoplasmic leaflet and replacing the single chains from the two chains of PG2 in the antiparallel dimer (Supplementary Fig. 6b, c, e).

Finally, in the antiparallel dimer, the PG3 density lies between linking transmembrane domain TM7 and TM8 that connect the two 6-helix bundles, and TM2, TM13 (Fig. 1e and Supplementary Figs. 8c and 9). The head group of PG3 is at the center of the central cavity with acyl chains extending out towards the extracellular leaflet (Fig. 1e). In the parallel dimer, PG3 fatty acyl chains were only weakly defined and were not interpreted.

In order to test the selectivity in the lipid binding sites we performed molecular dynamic (MD) simulations to examine the binding and stability of PG molecules (PG1, PG2 and PG3) in EfpA for a duration of 500 ns using three separate production MD runs. PG1 and PG3 molecules which are surrounded by protein exhibit a trajectory RMSD of <5 Å while the PG2 molecule in which one fatty acyl chain is exposed to the surrounding micelle exhibits more motion of that chain with a higher RMSD of <7 Å (Supplementary Fig. 9b). This is consistent with the tunnels representing stable productive binding sites for lipids within EfpA as seen in our structures.

The presence of PG in the recombinant EfpA protein suggests a pathway for lipid transport that could draw PG or similar lipids from the inner leaflet of the *Mtb* plasma membrane and deliver them to the outer leaflet or the mycobacterial periplasmic space[13]. To gain additional insight, we examined available, expanded PROSPECT data for the BRD-8000 series to define genetic interactions with EfpA inhibition. Specifically, when we examined the chemical genetic interaction profile for a potent analogue, BRD-7158 (MIC$_{90}$ = 0.39 μM; Fig. 1f), against an expanded set of 361 strains, each hypomorphic for a different essential target protein, we found strong sensitization of additional strains to BRD-7158, beyond EfpA, that are linked to PG synthesis. These highly sensitized strains included Rv2812c (PlsM), an essential acyltransferase, and CoaE, a dephospho-CoA kinase (Fig. 1f). PlsM is an early enzyme in phospholipid synthesis that transfers acyl-CoAs to the *sn-2* position of glycerol-3-phosphate[14] and *sn-1*-lysophosphatidic acid en route to the synthesis phosphatidic acid (PA) (Fig. 1f), while CoaE executes the final step in coenzyme A synthesis, which is required for the CoA-based activation of the acyl substrates of PlsM[15]. PA is subsequently used for the synthesis of phospholipids including PG, cardiolipin, phosphatidylinositol, and phosphatidylserine[16]. Thus, chemical genetic interactions between EfpA inhibitors and PG synthesis genes support the concept that EfpA may be involved in transporting PG from the inner to outer leaflet of the plasma membrane.

## BRD-8000.3 mechanism of inhibition

We confirmed that BRD-8000.3 binds to EfpA$^{EM}$ using ligand detected proton NMR, with an observed dissociation constant of 179 ± 32 nM (Fig. 2a). The cryo-EM structure of EfpA$^{EM}$ with BRD-8000.3 bound was determined to 3.45 Å resolution (Supplementary Fig. 10). BRD-8000.3 binds in a tunnel (tunnel 2) in EfpA$^{EM}$ that is close to the center of the bilayer (Fig. 2b, c and Supplementary Fig. 11). It displaces PG2 from its position in apo-EfpA with minimal distortion of the surrounding site (Fig. 2d). BRD-8000.3 inserted between TM9, TM11, TM12, and TM14 with its long axis parallel to the membrane plane (Fig. 2c). The BRD-8000.3 binding site shares many of the same residues as the binding site of one fatty acid chain of PG2 and is comprised of hydrophobic amino acid side chains that include A316$^{TM9}$, V319$^{TM9}$, M320$^{TM9}$, T373$^{TM11}$, I374$^{TM11}$, G377$^{TM11}$, L380$^{TM11}$, F381$^{TM11}$ I412$^{TM12}$, A415$^{TM12}$, G413$^{TM12}$, V416$^{TM12}$, L419$^{TM12}$, A498$^{TM14}$, I499$^{TM14}$, V501$^{TM14}$, and G502$^{TM14}$ (Fig. 2e).

Two glycine residues (G377$^{TM11}$ and G502$^{TM14}$) from TM11 and TM14 flank the BRD-8000.3 binding site in tunnel 2 between helices allow access from the bilayer (Fig. 2e center panel). BRD-8000.3's known resistance-conferring mutations (V319F or A415V) create steric hinderance to prevent access of BRD-8000.3 to its binding site. To

validate this further, we overexpressed WT and mutant EfpA alleles in *Mycobacterium bovis* BCG, an attenuated member of the *Mtb* complex. As expected, overexpression of EfpA mutant V319F resulted in a 13-fold increase in MIC$_{90}$ to BRD-8000.3. However, overexpression of an EfpA allele wherein V319 was changed to a smaller alanine (V319A) had no impact on BRD-8000.3 sensitivity, suggesting that the valine to alanine substitution still allows for inhibitor access and entry to its binding pocket (Fig. 2f).

The structure rationalizes aspects of the structure-activity relationship identified during optimization of the BRD-8000 series as a potent of how it might assist in structure-based optimization of compound affinity. Substitution of the 3-bromo pyridine in BRD-8000.1 for 2-bromo-pyridine in BRD-8000.2 places the bromine substituent into a hydrophobic pocket on just one side of the bromo-pyridine ring and produces a ~4-fold advantage in MIC$_{90}^{2}$. The further addition of a 1,3-pyrimidine at the 3- position of the pyridine allows extension of the long axis of the molecule further in toward the inside of hydrophobic tunnel 2 and provides a further ~3-fold advantage in MIC$_{90}$. This direction provides space and hydrophobic structure to support substitution. Resolution of the two trans-stereoisomers of the dimethyl vinyl group in BRD-8000 produced ~10-fold lower MIC$_{90}$ for the *(S,S)-trans* stereoisomer (BRD-8000.1) than its enantiomer. The active stereoisomer allows favorable hydrophobic interactions with I499, F38, Y378, I374 on the lipid-facing outside of the transporter where this asymmetric environment encodes stereospecificity.

## BRD-9327 mechanism of inhibition

BRD-9327 is a second EfpA inhibitor identified based on similarity of its chemical genetic interaction profile to that of BRD-8000.3[3] *M. marinum*, a close phylogenetic relative of *Mtb*, mutations that confer resistance to BRD-8000.3 do not confer resistance to BRD-9327 suggesting a different mechanism of inhibition[3]. The structure of BRD-9327-bound EfpA$^{EM}$ at 3.0 Å resolution showed density for BRD-9327 on the inside of the extracellular vestibule with its long axis parallel to transmembrane domains between TM1,TM5,TM9 and TM10 (Fig. 3a, b, Supplementary Figs. 12 and 13a, b). The binding pocket is comprised of T70$^{TM1}$, I73$^{TM1}$, V74$^{TM1}$, S191$^{TM5}$, V192$^{TM5}$, L195$^{TM5}$, T324$^{TM9}$, V325$^{TM9}$, I327$^{TM9}$, G328$^{TM9}$, L329$^{TM9}$, V331$^{TM9}$, Q332$^{TM9}$, S338$^{TM10}$, A339$^{TM10}$, A342$^{TM10}$, F346$^{TM10}$ and F349$^{TM10}$ residues (Fig. 3c).

Mutations identified to confer resistance to BRD-9327 in *M. marinum* include A339T, G328C, G328D and F346L[3] (numbering as homologous residue in *Mtb*). These residues are each part of the BRD-9327 binding pocket. Mutations in A339T, G328C and G328D appear to directly interfere with BRD-9327 binding, likely accounting for resistance to the inhibitor. Our MD simulations demonstrate that the phenyl ring of F346$^{TM10}$ partially forms π–π interactions with the benzyl bromide of BRD-9327 (Supplementary Fig. 14a, b); a leucine substitution (F346L) which confers resistance would abrogate this interaction. The side chain of Q332$^{TM9}$ above the compound is displaced upward in the presence of BRD-9327 resulting in shift of 2.6 Å from the apo state (Supplementary Fig. 13e, f). The head group of PG1 below the molecule and lower in the vestibule is displaced downward away from the compound (Supplementary Fig. 13d, e). The PG1 density is better resolved than in the apo structure, arguably consistent with the closer packing against the compound (Supplementary Fig. 15). The location of BRD-9327 suggests that it will partially block transport of a substrate if it were to pass through the tunnel (Fig. 3c) toward the cytoplasmic side. Also, it will inhibit the dynamical motion required to complete the alternating access transport cycle.

## Discussion

*Mtb* has 30 transporters that are members of the MFS superfamily[13], only one of which, EfpA, is essential[17]. This is the first structure of any of these transporters from *Mtb*. Interest in EfpA has predominantly centered around its role in drug resistance, with its upregulation in many

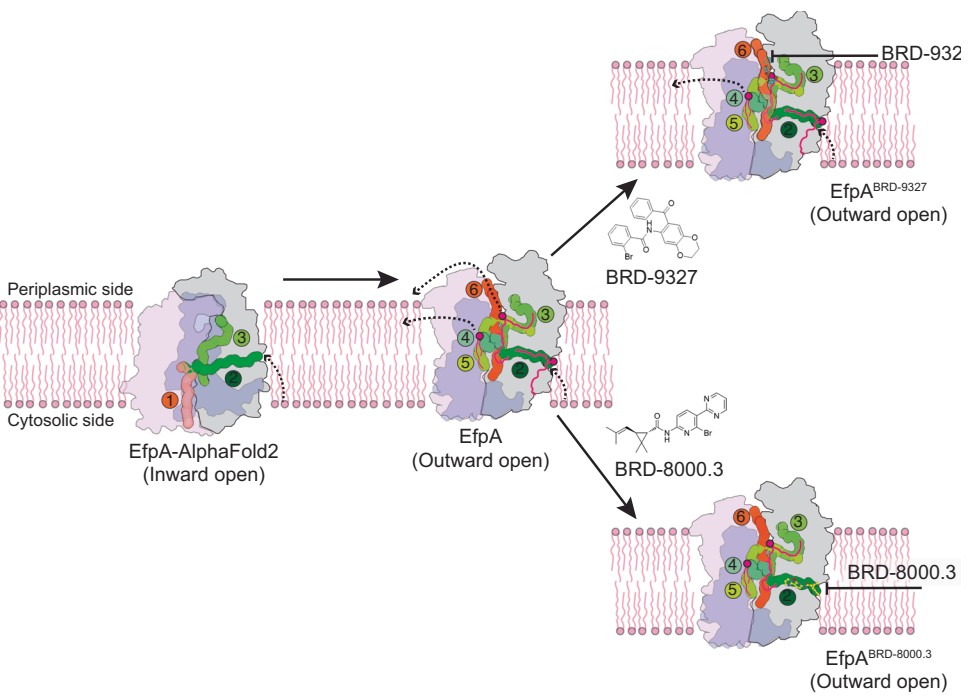

**Fig. 4 | Model of lipid transport by and inhibition of EfpA.** The inward facing state was predicted by AlphaFold2. One fatty acid chain of a lipid could bind in tunnel 2, transfer to tunnel 3 for release to the outer leaflet via either tunnel 6 or laterally to the outer leaflet via the 4/5 opening as seen for PG3 in the structure. Tunnels were defined by the MOLEonline server in the inward open model and outward open structure, labeled 1, 2, 3, 4, 5, and 6. Schematic showing the binding sites of BRD-8000.3 (bottom) and BRD-9327 (top) to the structure. BRD-8000.3 could be competitive with and block access of the substrate into or from the lipids. BRD-9327 binds in the extracellular vestibule and may interfere with dynamics of the alternate access mechanism, or possibly block export of the true substrate through the periplasmic vestibule in *Mtb*.

drug-resistant clinical isolates of *Mtb*[18] and evidence for its ability to efflux a number of first and second line *Mtb* drugs[19]. Meanwhile, the demonstration of its essentiality in the absence of antibiotics both genetically[20–22] and chemically, with our recent reports of EfpA small molecule inhibitors[2,3], points to a different, critical role for this transporter in bacterial survival and growth. This essential function of EfpA in *Mtb* however has not yet been elucidated, with limited characterization of EfpA outside of the context of drug resistance. EfpA depletion in *Mtb* using CRISPRi has been reported to cause changes in bacterial morphology, possibly suggesting a role in maintaining the cell wall[23].

Here we sought to determine the structure of EfpA and then to understand the mode of inhibition of two small molecule inhibitors, BRD-8000.3 and BRD-9327 that are not only synergistic, but display collateral sensitivity as resistance to one, confers sensitization to the other[3]. Using a BRIL-based strategy to determine the cryo-EM structure of EfpA, we found that its 14 TM helices are organized in two 6 -helix bundles each (TM1-TM6 and TM9-TM14), arranged with a pseudo twofold symmetry, as is typical for MFS transporters, with a 2TM insertion in the middle of the protein. Structure comparison of the outward open state with other QacA (quaternary ammonium compound A) family members including the *Staphylococcus aureus* QacA transporter shows that EfpA TM arrangement is very similar to QacA (Supplementary Fig. 16). The TM1-6 and TM9-14 of EfpA almost overlap with the QacA transporter while the TM7 TM8 pair are peripheral to the main helical bundles. In QacA[24,] TM7 TM8 are juxtaposed to space between TM2 and 13 and block the lateral opening while in EfpA TM7 TM8 are arranged opposite, towards TM2, in way that maintains the lateral opening (Supplementary Fig. 16b). This lateral opening is comprised of tunnel 4 and 5 opening into the outer leaflet where a third PG molecule is situated in EfpA (Fig. 4). The lipid transporter MFSD2A also has a lateral opening as in EfpA (Supplementary Fig. 16) where presence of lysophosphatidylcholine has been confirmed in the

MFSD2A structure[25]. The presence of this wide lateral opening in EfpA that could accommodate large lipid-like molecules, similar to the opening in MFSD2A, suggests that a lipid could be a native substrate for EfpA.

Long aliphatic chains of lipids occupy the tunnels formed within the recombinant protein isolated from *E. coli*. Identification of PG as one of the dominant co-purified lipid raises the interesting possibility that EfpA may be involved in lipid transport (Fig. 4), possibly for phoshatidylglycerol (PG), a lipid found in all kingdoms of life[26]. In support of this possibility in *Mtb*, we find strong chemical-genetic interactions between genes involved in PG synthesis and EfpA inhibitors, in particular the essential acyltransferase PlsM (Rv2812c) required for the production of phosphatidic acid, a precursor to phospholipids including PG, cardiolipin, phosphatidylinositol and phosphatidylserine[27] (Fig. 1f). Based on these data, we propose that EfpA could play a role in transport of PG or similar lipid.

The polar phosphatidyl head groups of PG1, PG3 lie within the outward-facing vestibule and accessible to solvent, with some small polar side chains in the vicinity though no counterion per se. PG2 and PG2' present their two head groups close together in a cleft formed on the outer surface of the dimer interface between two EfpA molecules, buttressed between TM14 and TM14'. This location is unusual in that it does not provide a polar environment or any counterion in this environment within the LMNG micelle. While the parallel dimer we see could perhaps facilitate the association there. that the antiparallel dimer is a non-physiological arrangement. However, one of the fatty acid chains occupies tunnel 2 in both cases and is supported by an almost uniformly hydrophobic surrounding. This suggests that the energetics of binding in tunnel 2 may support the energetic cost of entry of the phosphatidyl head group into the lipidic leaflet of the bilayer. From here it could move into the protein domain to positions PG1 or PG3.

After we had completed these structures in 2023 a preprint publication of Wang et al. in 2024 reported an EfpA structure at 3.1 Å from

protein expressed in Expi293 cells[28,29]. They also found that MtEfpA formed an antiparallel dimer and contained fatty acid chains of lipids at similar positions to PG1, PG2, PG3. While the fatty acid aliphatic chains are well defined, the head groups in neither of our cases are uniquely defined in the density maps, and the lipids bound in each case were derived from the different expression systems so they may be different, and different from the lipids or true substrates in *Mtb*. Their mass spectrometry identified lipids associated with their sample as PE, PG, and CDL. While mass spectrometry is not quantitative as to relative amounts in the sample, based on preponderance of PE by mass spectrometry they interpreted their density as the presence of two PE (rather than PG) and one CDL that spans between two EfpA molecules in the antiparallel dimer. The size of the head group in PE is two heavy atoms smaller than in PG with the same backbone between fatty acid chains, hence seeing the difference by cryo-EM would be unlikely at these resolutions. PG is a precursor of CDL that resembles a dimer of PG. The density in both our structures resembles either a dimer of PG molecules, or one CDL that spans two monomers in the antiparallel protein arrangement.

To question whether the asymmetric dimer was the physiological form of EfpA, Wang et al. showed that a GFP tag fused to the C-terminus of EfpA was only exposed on the inside of the expressing HEK cell arguing that the asymmetric dimer was not present in HEK cells and might be something created during purification in detergent. Furthermore, they also expressed EfpA from *M. smegmatis* in HEK cells and found it purified as a monomer in a detergent mixture of DDM/CHS. A second recent paper by Li et al. [30] reported the structure of EfpA expressed in *Schizosaccharomyces pombe* (*S. pombe*), and in *M. smegmatis* that is also purified as an antiparallel dimer, and argue that the antiparallel dimer may be physiological and functional in contradiction of the GFP tagged result. If so the addition of the GFP tag of Wang et al. might possibly have disfavored translocation of the GFP across the membrane[28,29]. The structure of Li et al. identified two (rather than three) lipids that correspond to PG1, and the two penetrant halves of PG2 that therefore are interpreted as one PA lipid with one fatty acid chain penetrating each monomer of the asymmetric dimer[30]. In each case and our own structures, the lipid that intersects the twofold axis between molecules (at the PG2 site) would place one, or two negatively charged phosphates in the case of CL, outside the dimer interface and in contact with the micelle. Adding further to the issue, the structure of *M. smegmatis* MsEfpA as a monomer showed no density for lipid at the PG2 site.

The finding that MsEfpA does not dimerize[28] raises the question as to why since the biology of *M. smegmatis* and *Mtb* are similar, and the sequence of MsEfpA is 77% identical to MtbEfpA. The primary residues in the dimer interface however are different between *Mtb*/ *M. smegmatis* and are substituted as follows Y378I, L385I, I374I, I499V (Supplementary Figs. 5 and 6) which could explain the lack of dimerization in *M. smegmatis*. Hence the function, if it depended on dimerization would be different in *M. smegmatis*, perhaps arguing against a functional role for antiparallel dimerization.

In our case we also found and determined the structure of a parallel dimer. The dimer interface is based on the same residues as in the antiparallel dimer with the interacting side chains being left to right in the parallel dimer (Supplementary Fig. 6), versus top to bottom as in Supplementary Fig. 5.

These structures all suggest that certain yet undefined lipids from *Mtb* are the substrates for EfpA, with differences in suggested assignment depending on the three different expression systems from which the protein and lipids were derived. The antiparallel dimer was also postulated by Li et al., to correspond to the physiological form of EfpA. While antiparallel dimers of certain transporters in the much smaller EmrE class are recognized to function as antiparallel dimers[31–34], there are few if any SLC transporters that are known to function this way. If the membrane insertion mechanism allows for either orientation it

might allow for a possible mechanism of lipid translocation that might depend on rotation from inward facing to outward facing in the membrane plane.

Our BRD-9327 bound structure shows that it binds to one side of the outer-facing vestibule. It slightly displaces one fatty acid chain of PG1 site but does not prevent its binding. It binds close to three residues that were previously identified by the Hung group using mutational analysis. Our interpretation is that BRD-9327 may partially block substrate transport, and that it may serve to prevent closure of the outer gate. Hence it may be partially uncompetitive with lipidic substrates. Wang et al., carried out a focused docking experiment against the region of the residues that had been identified as affecting EtBr transport by us. The docking site was suggested to prevent PG1 site binding, however the docked pose is incorrect by 90° and impinges on, but does not prevent, lipid binding.

To propose a model of lipid transport, we used Alphafold2 to predict an inward open conformation of EfpA. While all of our cryo-EM structures are in an outward open conformation, we hypothesized the existence of an inward open conformation as is typical of MFS transporters which alternate between inward and outward open conformations as substrates are transported from one face to the other. Alphafold2 predicted another tunnel 1 from the cytoplasmic side (Fig. 4). Thus, substrates such as EtBr, which was the substrate used to evaluate EfpA inhibition[3] could enter via tunnel 1 while lipid molecules could enter EfpA from the interleaflet region of the plasma membrane via tunnel 1 or tunnel 2 in the inward open conformation. From tunnel 2, the lipid molecule could flop to tunnel 3 and ultimately be transported to the outer leaflet via opening of either tunnel 4 or 6 seen in the outward open structure (Fig. 4). In the gram-negative bacteria *E. coli*, there is a preponderance of PG in the outer leaflet of the membrane in comparison to the inner leaflet[35]. The arrangement of three PG molecules in the structures suggests that EfpA could play a crucial role in maintaining plasma membrane phospholipid asymmetry by transporting the PG from the inner to the outer leaflet (Supplementary Fig. 9a), where it could play a role in membrane organization or fluidity[36], or serve as a diacylglycerol donor for lipidation of lipoproteins[37,38]. This proposed role would also then be consistent with the morphological changes observed in *Mtb* with knockdown of EfpA[23].

The inhibitors were uncompetitive with ethidium bromide (EtBr) efflux by EfpA. As they do not obviously block entry at tunnel 1, the presumed entry point of EtBr, or exit from tunnel 6 on the periplasmic side of the membrane, they might achieve this by inhibiting the dynamic transition from inward to outward facing EfpA. This could be the mechanism by which BRD-8000.3 and BRD-9327 inhibit efflux of antitubercular drugs. In contrast, we might expect that BRD-8000.3 has two different ways by which it could inhibit EfpA transport of a natural lipid substrate. As BRD-8000.3 binds in hydrophobic tunnel 2 and displaces one fatty acid chain of the more interior of the PG molecules (PG2) found in the apo structure, it could be competitive with lipid substrates, excluding their entry into tunnel 2 from the lipid bilayer. Additionally, inhibiting the dynamic motions of TM9, TM11, TM12, TM14 could also be key features of the mechanism of BRD-8000.3 inhibition of efflux of a lipid substrate, in a similar manner as that proposed for the small molecule EtBr substrate. In the case of BRD-9327 which binds inside the external vestibule near the external opening of tunnel 6 (Fig. 4). it is uncompetitive with EtBr binding, and does not alter the protein conformation, so does not seem to act as an allosteric inhibitor. Instead, its mode of action could be by blocking either the dynamics of the alternate access mechanism to switch between inner and outer conformations for both natural and unnatural substrates, and/or impeding expulsion of big, natural substrates (larger than EtBr) from the exit portion of the efflux pathway.

In summary, the structures we present here uncover different modes of inhibition for inhibitors (BRD-8000.3 and BRD-9327) and support the idea that compounds sharing the same target could be

used in combinatorial drug design to treat tuberculosis, potentially with favorable implications for the emergence of resistance[3]. To solidify the hypothesis of possible combinatorial action of these inhibitors, we determined structure of EfpA with both BRD-8000.3 and BRD-9327 exhibiting binding of each at two different locations simultaneously in each monomer transporter (Supplementary Fig. 17). They also provide a crucial platform for the structure-based modification of these compounds to further improve their efficacy. As EfpA is an essential efflux pump in *Mtb*, drugs that inhibit EfpA can inhibit growth of *Mtb* while simultaneously enhancing the activity of companion drugs by limiting their efflux[5,19], thereby making EfpA an attractive target. Finally, these structures also provide insight into possible functions of EfpA, thereby demonstrating that chemical tools such as these inhibitors can play valuable roles in helping to elucidate the crucial, biological functions of essential targets.

## Methods

### Protein expression and purification

The *Mycobacterium tuberculosis* EfpA (Efflux protein A) protein (Uniprot: P9WJY4) encoding gene along with N-terminal Flag (DYKDDDK) epitope tag and C-terminal 10X histidine-tagged fusion protein was codon-optimized and cloned into the pET-28a(+) bacterial expression vector (Twist bioscience). The BRIL-tag construct was prepared from it by replacing the first 48 amino acids of EfpA that are predicted to be disordered by Alphafold2 and lie in the cytoplasmic space with BRIL (a variant of apocytochrome b562)[7,39,40] to serve as a fiducial marker. The final EfpA-EM construct was derived from BRIL-EfpA by mutating proline at position 171 to arginine (GenScript) to provide possible interactions that might help order the BRIL domain.

The resulting construct pET-28a-BRIL-P171R_EfpA was transformed into *E. coli* BL21(DE3) strain (NEB) and a primary culture was started from a single colony obtained after transformation. From the primary culture a secondary culture was set up in 4 flasks with 1 L of TB media each with 50 μg/mL kanamycin and grown at 37 °C, 200 rpm up to an optical density ($OD_{600nm}$) between 0.6 and 0.8. Induction was done by adding 0.5 mM of IPTG and grown for another 4 h 37 °C, followed by harvesting using centrifugation and stored at −80 °C.

For protein purification, cells were resuspended in ice cold lysis buffer (50 mM Tris-Cl pH 8.0, 500 mM NaCl, DNase I and cOmplete, EDTA-free protease inhibitor cocktail tablets (Roche)) and lysed using sonication pulse (1 s on and 1 s off) on ice for 3 min for 5 cycles. The lysate was spun at $112,967 \times g$ for 2 h ultracentrifugation to separate the crude membrane and membrane pellet and stored at −80 °C. Detergent extraction of protein was performed by resuspension of membrane (15 mL buffer/1 g membrane) in ice cold resuspension buffer (50 mM Tris-Cl pH 7.5, 300 mM NaCl, 0.5% 2,2-didecylpropane-1,3-bis-β-D-maltopyranoside (LMNG)/0.05% cholesteryl hemisuccinate (CHS) and protease inhibitor) at 4 °C overnight. Insoluble material was separated by centrifugation at $34,155 \times g$ for 30 min at 4 °C and supernatant was filtered through a 0.4 μM filter.

Filtered detergent-solubilized protein was supplemented with 30 mM imidazole pH 7.5 and was loaded onto a 5 mL HisTrap HP column (Cytiva) at a flow rate of 1.5 mL/min. The column was washed extensively first with Buffer A (50 mM Tris-Cl pH 7.5, 300 mM NaCl, 0.01% LMNG/0.001% CHS) followed by Buffer A supplemented with a gradient of imidazole, concentration (30 mM, 50 mM, 80 mM, 120 mM). Protein was eluted in 20 mL elution buffer (50 mM Tris-Cl pH 7.5, 300 mM NaCl, 300 mM imidazole, 0.01% LMNG/0.001% CHS). Protein was concentrated using a 50-kDa filter (Amicon) and further purified by injecting into a Superdex 200 Increase 10/300 GL column (Cytiva) equilibrated with buffer A. Fractions containing EfpA protein were concentrated, flash frozen in liquid nitrogen, and stored at −80 °C.

### Anti-BRIL Fab (BAG2 Fab) purification

The BAG2 Fab expression construct in pRH2.2 vector was a gift from Kossiakoff[8]. This vector was transformed into competent C43(DE3) *E. coli* cells (Sigma-Aldrich) and grown overnight in 10 mL 2xYT supplemented with 100 μg/mL ampicillin shaking at 200 rpm and 37 °C. 10 mL overnight culture was used to start secondary culture in 1 L of sterile TB autoinduction media (Terrific Broth supplemented with 0.4% glycerol, 0.01% glucose, 0.02% lactose and 1.25 mM $MgSO_4$ and 100 μg/mL ampicillin) in a 2.8 L baffled flasks at 37 °C for about 6 h and then decrease the temp to 30 °C and leave it overnight at 225 rpm. The next day, cells were harvested by centrifugation and stored in the −80 °C. Purification was performed by lysis of cells using sonication in 50 mM Tris-Cl pH 7.5, 300 mM NaCl supplemented with protease inhibitors, DNase I, and the soluble fraction was heated in a 60 °C water bath for 30 min. High-speed centrifugation was performed to remove aggregates and the supernatant affinity purified using protein L purification (Cytiva) with the manufacturer's protocol. The eluted Fab was concentrated with a 10 kDa MWCO spin and further purified with size-exclusion chromatography, using a Superdex S75 Increase 10/300 column (Cytiva) equilibrated in 50 mM Tris-Cl pH 7.5, 300 mM NaCl and purified sample was flash frozen in liquid $N_2$ and stored in −80 °C.

### Expression and purification of anti-Fab elbow Nb

DNA encoding Nb[41] was cloned into a pET-26b expression vector with N-terminal 6x His and Nb sequence. The expression vector was transformed into BL21(DE3) cells (NEB), and primary culture started in 50 mL 2xYT. A secondary culture was started in 6 flasks with 1 L 2xYT each with 50 μg/mL kanamycin and grown at 37 °C until mid-log ($OD_{600n}$-0.6–0.8) phase and induced with 1 mM IPTG. Expression was carried out overnight at 25 °C. Culture was pelleted down and lysed by sonication in 100 mL lysis buffer (50 mM Tris, 500 mM NaCl, protease inhibitors, and DNase I). The lysate was spun down at $20,000 \times g$ for 45 min and supernatant filtered using a 0.4 μM filter. The sample was supplemented with 5 mM imidazole and loaded onto 5 mL HisTrap HP column (Cytiva). The protein was purified from the Ni column by the protocol used for the EfpA purification. Purified protein sample were concentrated up to 5 mL using a 3 K centrifugal filter and further purified using a Superdex S75 Increase 10/300 column (Cytiva), equilibrated in 50 mM Tris-Cl pH 7.5, 300 mM NaCl and stored in −80 °C.

### Cryo-EM sample preparation and data collection

For cryo-EM sample preparation, the purified EfpA-EM protein, anti-BRIL Fab, and anti-Fab elbow Nb were mixed in the ratio of 1:1.5:2 and incubated for 1 h on ice. After incubation, the complex was separated using a Superose 6 Increase 10/300 GL column equilibrated with 50 mM Tris-Cl pH 7.5, 300 mM NaCl, and 0.02% GDN. The peak fractions corresponding to the ternary complex of EfpA-EM protein with Fab and Nb were collected (Supplementary Fig. 1)

For EfpA apo structure, 4 μL of protein complex sample at concentration of 16.16 mg/mL was applied to cryo-EM grids Quantifoil Au R1.2/1.3, 400 mesh (Electron Microscopy Sciences) using Mark IV Vitrobot (FEI) at 8 °C, 100% humidity and blotted for 3.5 s followed by plunge-freezing in liquid ethane.

A total of 16,938 movies were recorded for EfpA apo samples on a Titan Krios at 300 kV equipped with a K3 Summit detector (Gatan) using SerialEM v4.1 beta software at UCSF. Data were collected in non-super resolution mode at 105,000X magnification (pixel size of 0.835 Å/pix) with a defocus range of -1 to -2 μm. Each movie contains 80 frames with a total electron dose of ~45 electrons /Å². 

For BRD-8000.3 sample the protein was incubated with BRD-8000.3 100 μM final concentration on ice for 15 min and 4 μL of sample of final 5.91 mg /mL protein with 100 μM BRD-8000.3 was applied on cryo-EM grids Quantifoil Au R1.2/1.3, 400 mesh (Electron Microscopy Sciences) before being plunged into liquid ethane equilibrated to -185 °C.

A total of 10,235 movies were recorded EfpA-BRD-8000.3 samples on a Titan Krios at 300 kV equipped with a K3 Summit detector (Gatan) using SerialEM v4.1 beta software at UCSF. Data were collected in super resolution mode at 105,000X magnification (pixel size of 0.4175 Å/pix) with a defocus range of −1 to −2 μm. Each movie contains 80 frames with a total electron dose of ~45 electrons /Å$^2$.

For BRD-9327 bound sample, protein was incubated with 200 μM final concentration of BRD-9327 on ice for 15 min and 4 μL of sample of final 8.9 mg /mL protein with 200 mM BRD-9327 was applied on cryo-EM grids Quantifoil Au R1.2/1.3, 400 mesh (Electron Microscopy Sciences) before being plunged into liquid ethane equilibrated to -185 °C. A total of 12,748 movies were recorded on a Titan Krios at 300 kV equipped with a K3 Summit detector (Gatan) using SerialEM v4.1 beta software at UCSF. Data were collected in super-resolution mode at 105,000X magnification with physical pixel size of 0.417 Å/pix and defocus range of -1 to -2 μm. Each movie contains 80 frames with a total electron dose of ~43 electrons /Å$^2$.

### Cryo-EM data processing

Raw movies were motion-corrected using UCSF MotionCor2v1.4.1[42]. For BRD-8000.3 and BRD-9327 super resolution mode motion-corrected data to Fourier bin images 2 × 2 to the counting pixel size 0.835 Å/pix and 0.834 Å/pix respectively. Dose-weighted micrographs were imported into cryoSPARC[43] v4.2.1 and split in two to three subset as shown in supplementary data (Supplementary Figs. 2, 10 and 12) for faster data processing and to regulate the computational memory issue during data processing. The defocus values on these subsets were estimated using patch CTF estimation (multi-GPU). Particle picking, extraction, classification and refinement were performed in cryoSPARC v4.2.1.

Initially blob picking was performed with minimum and maximum particle diameter 100 Å and 180 Å respectively on a subset of micrographs as shown in Supplementary Figs. 2, 10 and 12. The particle extraction was performed with a box size of 440 pixel at 4X binned, and 2D classification was performed. Classes from 2D classification were select by visual inspection and used as templates for the automated particle picking. The particles were extracted with box size 440 at 4X binned and subject to multiple rounds of 2D classification followed by ab-initio 3D map generation as shown in Supplementary Fig. 2, 10 and 12. Ab-initio maps were used as the reference template and heterogenous refinement was performed. After heterogenous refinement based on visual inspection the classes were selected, and un-binned particles were re-extracted. Each class particles were subjected to non-uniform refinement using the respective map obtained from heterogenous refinement (Supplementary Figs. 2, 10 and 12). After non-uniform refinement a focused mask around the trans-membrane domain was generated in cryoSPARC volume tools with dilation radius 6 and soft padding width 14. Respective masks for each class have been shown in Supplementary Figs. 2, 10 and 12. Finally the mask is used for local refinement and final map is obtained for each data as shown in Supplementary Figs. 2, 10 and 12. A summary of the data processing and final EM map quality is presented in Supplementary Figs. 2, 10 and 12.

### Model building and refinement

A model of EfpA from AlphaFold2 (https://www.uniprot.org/uniprotkb/P9WJY5/entry#structure) was used as starting model and rigid body and manual fitting was performed in COOT[44] followed by real-space refinement using Phenix[45,46] against the final map. BRD-8000.3, BRD-9327, PG and CHS restraints were generated using Phenix eLBOW. The final model with ligands was obtained by real-space refinement using reference models and Ramachandran restraints for both inhibitors. The final model statics are reported in Supplementary Table 1. Figures were prepared using UCSF ChimeraX[47,48].

### MIC assays

The efpA coding region from *M. tuberculosis* H37Rv was PCR amplified using primers efpA_pUV_clone_F and efpA-pUV-R (Supplementary Table 2) synthesize from IDT and cloned into the PCR4 TOPO vector using the Invitrogen TOPO TA cloning kit (REF 45-0071). V319F and V319A mutations were introduced using Agilent QuikChange II Site-Directed Mutagenesis Kit (Cat #200524) with primers mentioned in Supplementary Table 2. WT and mutant *efpA* regions were excised from the TOPO vector using SspI and PacI and cloned into the myco-bacterial shuttle vector pUV15tetORm digested with PacI and EcoRV. The resulting constructs were transformed in *Mycobacterium bovis* BCG (Pasteur).

BCG was grown in Middlebrook 7H9 medium (Difco) supplemented with 10% OADC (BBL), 0.2% glycerol, 0.05% tyloxapol, and 50 μg/ml hygromycin B throughout the experiment. Transformants were grown to mid log phase ($OD_{600nm}$ 0.2–0.3) and EfpA over-expression was induced 1 d prior to MIC assay with 100 ng/ml anhy-drotetracycline (AHT). Following overnight induction, cultures were back diluted to $OD_{600nm}$ 0.0025 prior to transfer into 96 well assay plates (100 μl/well) containing 1 μl/well of BRD-8000.3 serially diluted in DMSO. Plates contained 3 replicates at each dilution and were incubated at 37 °C for 10 days.

### EtBr efflux assay

Wild type EfpA and EfpA$^{EM}$ genes were cloned into a pBAD vector and transformed into JD838 cells, a strain derived from *E. coli* K-12 in which three drug transporters are deleted (genotype: *ΔmdfAΔacrBΔydhE::kan*). The JD838 strain has been previously used for EtBr transport assay with different MFS transporters[49]. The bacterial cultures were grown at 37 °C in Luria Bertani (LB) media to an $OD_{600nm}$ of 0.4 before induction of EfpA expression with 0.05% arabinose. After 2 h, the cells were harvested, washed with PBS, and the pellet resuspended in EtBr-free PBS by adjusting the $OD_{600nm}$ to 1. The cells were then loaded with 20 μM Ethidium Bromide (EtBr) in the presence of 0.5 μM of carbonyl cyanide m-chlorophenyl hydrazone (CCCP) and incubated in the dark at 37 °C for 45 min. After incubation, the cells were washed three times and resuspended in PBS again by adjusting the $OD_{600nm}$ to 0.8. EtBr efflux was initiated by adding 50 μL of cells into a 96-well-clear bottom plate containing 50 μL 0.8 % glucose in PBS. EtBr fluorescence was then monitored continuously for 45 min using a SpectraMax M5 plate reader (Molecular Devices) preset to an excitation wavelength of λ = 530 nm and emission wavelength λ = 585 nm.

### Extraction of lipids co-purified with EfpA

Lipids co-purified with the protein sample were extracted using the method of Bligh and Dyer as described[50]. Briefly, the protein solution (containing ~1 mg protein) was transferred into a glass tube, and phosphate-buffered saline (PBS) solution was added to a final volume of 1.6 mL. Then, 2 mL of chloroform and 4 mL of methanol were added to make the single-phase Bligh-Dyer mixture, which consists of chloroform/methanol/PBS (1:2:0.8, v/v/v). This solution was subjected to sonic irradiation in a bath apparatus for 5 min. This single-phase extraction mixture was then centrifuged at $500 \times g$ for 10 min in a clinical centrifuge to pellet the protein precipitate. The supernatant was then transferred to a fresh glass tube where 2 mL of chloroform and 2 mL of PBS were added to generate the two-phase Bligh-Dyer mixture, which consists of chloroform/methanol/PBS (2:2:1.8, v/v/v). After mixing and centrifugation ($500 \times g$) for 10 min, the upper phase was removed, and the lower phase was dried under a stream of nitrogen. The dried lipid extract was stored at −20 °C until LC/MS analysis.

### Lipid identification by LC/MS/MS

Normal phase LC was performed on an Agilent 1200 Quaternary LC system equipped with an Ascentis Silica HPLC column, 5 μm,

25 cm × 2.1 mm (Sigma-Aldrich) as previously described[51]. Mobile phase A consisted of chloroform/methanol/aqueous ammonium hydroxide (800:195:5, v/v/v); mobile phase B consisted of chloroform/methanol/water/aqueous ammonium hydroxide (600:340:50:5, v/v/v); mobile phase C consisted of chloroform/methanol/water/aqueous ammonium hydroxide (450:450:95:5, v/v/v). The elution program consisted of the following: 100% mobile phase A was held isocratically for 2 min and then linearly increased to 100% mobile phase B over 14 min and held at 100% B for 11 min. The LC gradient was then changed to 100% mobile phase C over 3 min and held at 100% C for 3 min, and finally returned to 100% A over 0.5 min and held at 100% A for 5 min. The LC eluent (with a total flow rate of 300 µm/min) was introduced into the ESI source of a high-resolution TripleTOF5600 mass spectrometer (Sciex). Instrumental settings for negative ion ESI and MS/MS analysis of lipid species were as follows: IS = −4500 V; CUR = 20 psi; GSI = 20 psi; DP = −55 V; and FP = −150 V. The MS/MS analysis used nitrogen as the collision gas. Data analysis was performed using Analyst TF1.5 software (Sciex). In view of other considerations (such as potentially non-specific protein-lipid binding), we didn't pursue rigorous lipid quantification. The assay was performed with one biological replicate.

## Molecular dynamics simulations

All atom molecular dynamics (MD) simulations were performed with the GROMACS simulation engine[52] using the CHARMM36 force field[53] parameters. The cryo-EM structure was used as the starting point for the MD simulations. CHARMM-GUI[54] was used to prepare the system by embedding the protein in a lipid bilayer which was modeled using POPC molecules. The initial membrane coordinates were designated by the PPM server through the CHARMM-GUI interface. The system was further solvated using the TIP3P water model with the inclusion of 150 mM of Na$^+$ and Cl$^-$ ions. The solvated system was minimized for 5000 steps using the steepest descent method. Furthermore, the system was equilibrated with default parameters supplied by CHARMM-GUI, starting with weak restraints, and successively reducing the strength of the restraints in 5 steps for a total time of 50 ns. Additionally, a 100 ns equilibration simulation was run without any restraints. Following this, production simulations were executed for 500 ns within the NPT ensemble, employing a Parrinello–Rahman barostat set to 1 atm and a V-rescale thermostat maintaining a temperature of 303.15° K. The SHAKE algorithm was applied to enforce constraints on all bond lengths involving hydrogens and the hydrogen mass repartitioning scheme was utilized for the production runs, enabling a time step of 4 fs. The topology, initial and final configurations, MD-simulation parameter, and result files of all three trajectories are provided in supplementary data 1.

## BRD-8000.3 binding assay

1H-NMR experiments were recorded on a Bruker 600 MHz spectrometer equipped with a cryogenic QCIF probe and an automatic sample handling system. Experiments were conducted either at 280 K or 298 K and processed using the spectrometer's TopSpin 3.6 software. Intensities of the most prominent aromatic proton peak ~8.9ppm were extracted in a titration series with and without protein and normalized to reflect free and bound BRD-8000.3. Samples typically contained 30 µM BRD-8000.3 in NMR buffer (50 mM Tris-d7 300 mM NaCl, 0.01% LMNG and .001%CHS at pH7). The data were fit to a standard quadratic binding model to determine the apparent affinity[55].

## PROSPECT of BRD-7158

PROSPECT was executed as described[2]. BRD-7158 was among 21 compounds with known targets profiled in 10-point dose response using an expanded 361-strain PROSPECT pool (354 hypomorphic strains and 7 barcoded H37Rv strains). After initial calculation of strain-specific log2-fold changes as described previously[2] strain-by-strain effects were converted to growth rate (GR, 1 = no effect, 0 = full effect) based on comparison to strain behavior in DMSO and rifampin (100 nM) respectively. GR for each strain was then standardized (z-scored) across all test treatments in the screening data, generating a standardized growth rate (sgr) metric that provides confidence in the scale of the measured effect.

## Reporting summary

Further information on research design is available in the Nature Portfolio Reporting Summary linked to this article.

## Data availability

The atomic coordinates for four structures of EfpA apo antiparallel dimer, parallel dimer, BRD-9327 bound and BRD-8000.3 bound have been deposited in the Protein Data Bank with the accession codes PDB ID 9BII (EfpA antiparallel dimer), 9BL7 (EfpA parallel dimer), 9BIQ (BRD-9327 bound EfpA) and 9BIN (BRD-8000.3 bound EfpA) respectively. The corresponding maps have been deposited in the Electron Microscopy Data Bank with the accession codes EMD-44591 (EfpA antiparallel dimer), EMD-44651 (EfpA parallel dimer), EMD-44598 (BRD-9327 bound EfpA) and EMD-44594 (BRD-8000.3 bound EfpA). Access codes for other published atomic coordinates used for comparison purposes were provided accordingly in the manuscript and figure legends, including: 7Y58 and 7N98. Source data are provided with this paper for Fig. 1f, Fig. 2a, f, Supplementary Fig. 1b–d. Source data are provided with this paper.

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

## Acknowledgements

The research was supported by the NIH National Institute of General Medical Sciences (NIGMS R01GM024485) to R.M.S. Research to Z.G.

was supported by NIH National Institute of Allergy and Infectious Diseases (NIAID R01AI178692). Research to D.T.H. was supported by the Bill and Melinda Gates Foundation. We thank Anthony Kossiakoff for provision of the anti-BRIL Fab, and nanobody against it. We thank Abhijit A. Sardesai for gift of the *E. coli* JD838 strain. We thank Sun Kyung Kim for help in expression cloning. We thank D. P. Bulkley, G. Gilbert and M. Harrington for support with cryo-EM data collected at the UCSF cryo-EM facility, which is supported by NIH grants S10OD020054, S10OD021741 and S10OD026881. We thank Janet Finer-Moore for critical reading of the manuscript.

## Author contributions

N.K.K., D.T.H. and R.M.S. conceived the project. N.K.K. and M.G. prepared the protein. M.G. and N.K.K. prepared EM grids, collected data and processed cryo-EM data. N.K.K. and M.G. performed model building and refinement. Z.G. determined the lipids by mass spectrometry. S.B. performed the NMR study. J.E.G. and A.Y.E. performed the PROSPECT analysis and MIC testing in BCG. T.K. performed chemical synthesis of compounds. S.G.B. and A.S. performed the MD-simulation analysis. All authors analyzed the results. N.K.K., M.G., J.E.G., D.T.H. and R.M.S. wrote the manuscript with contributions from all other authors.

## Competing interests

The authors declare no competing interests.
