## [Transparent Peer Review file · Nature Communications]

Structure and inhibition mechanisms of Mycobacterium tuberculosis essential transporter efflux protein A

Corresponding Author: Professor Robert Stroud

Version 0:

Reviewer comments:

Reviewer #1

(Remarks to the Author)

The authors of this manuscript evaluated the mechanism of inhibition of two inhibitors of the EfpA efflux pump of Mtb that were recently discovered by chemical-genetic screening, thereby exploring the previously uncharacterised biomolecules that appear to be transported across the membrane by the pump. It is clear from the literature that EfpA is associated with drug resistance and is required by Mtb, making it a perfect target for future studies. This makes it an interesting topic, especially given the severity of molecular mimicry by Mtb and the lack of suitable antibiotics. This review is based on my expertise in the field of lipidomics. In general, there is a lack of specificity in the evaluation of your lipidomics measurement. I would have expected an LC-MS measurement with subsequent data analysis of the peaks with MSMS fragmentation interpretation via software or manually. As this lipidomics measurement is part of your further explanation of how the inhibitor potentially blocks the lipid transport of the pump, the lipid data are an essential part.

Major comments:

- Even if it is only in the extended data, it is not sufficient to show a chromatogram as a result. It would be better to give areas instead.
- The method states that the lipids were measured in negative ionisation mode. Did you also check positive ionisation, as some lipids can only be measured in one mode and others in both?
- How many replicates were measured to conclude that PG is the most abundant lipid class? In another paper, currently on Researchsquare (10.21203/rs.3.rs-3740027/v1.), the authors hypothesise that there are not three PGs, but 2 PE and 1 cardiolipin binding at certain regions. As you have already discussed, 2 of the 3 PGs detected could also represent a cardiolipin. Since both your and the other manuscript state that the exact lipids responsible remain elusive, please briefly discuss the lipid combination presented in the other manuscript.
- In the chromatogram, since you have defined the peaks as lipid classes, what specific lipid species have you detected?
- Was a standard included? To normalise the data it would be nice to have a standard for each lipid class included in the measurement.

Minor comments:

- Since a QTOF was used for MS detection, was it run in scan mode only to detect all lipids? A subsequent MRM acquisition would have allowed additional quantification. This could also be considered in a future study.

Reviewer #2

(Remarks to the Author)

The manuscript by Khandelwal et al. reports the structures of Mycobacterium tuberculosis essential transporter efflux protein A (EfpA) with and without bound inhibitors. The structures obtained in the absence of the inhibitors reveal clear densities potentially corresponding to bound lipid molecules derived from the expression host. Based on the structural data along with the molecular dynamics simulations and lipidomic analysis, authors propose that the native substrate of EfpA could be phospholipids, specifically phosphatidylglycerol (PG). The structures in the presence of inhibitors BRD-8000.3 and BRD-9327 reveal the binding sites for these compounds through clear cryo-EM densities.

Overall, the structural data presented is strong and deserve publication in Nature Communications. While the manuscript contains substantial number of speculations that cannot readily be explained by the available data, these parts are discussed sufficiently and clear to the reader for the most part.

- The phosphate moiety of the proposed PG2 molecule would be facing to the hydrophobic core of the membrane in the absence of the dimeric arrangement that appears to be artificial. The authors should discuss this fact in the manuscript more in detail.
- The maps for the ligands should be shown with the maps of the surrounding residues to be able to judge the strength of the density.
- How was the alpha fold models were obtained. No specifics were included in the manuscript.
- Did the authors try C2 symmetry? The resolution of the maps appears sufficient, but it may be further improved with enforcing C2 symmetry and performing symmetry expansion to perform local refinement focusing on one of the protomers within the dimer.
- It would be interesting to see if arginine mutated in place of P171 forms salt bridges as initially expected.
- Cryo-EM data processing part should be described more in detail in the methods section. As it is, no specifics were given and the readers are referred to supplementary figures.
- Please clarify if the real space refinement was done together with the ligands. It is not clear in the methods section.

Reviewer #3

(Remarks to the Author)

In this manuscript, Kumar Khandelwal et al. resolved the structures of Mtb EfpA, a pharmacologically relevant multi-drug efflux pump, bound to the recently identified inhibitors BRD-8000.3 and BRD-9327 using single particle cryoEM. Identification of the BRD compounds was performed with the authors in-house developed PROSPECT pipeline (<https://doi.org/10.1021/acscinfecdis.9b00373>). Identification of the apo, and multiple holo conformations, i.e., BRD-8000.3, BRD-9327, and a combination of the two, provides insights as to the mechanism of the two non-competitive inhibitors of Mtb EfpA. Furthermore, with the addition of an AlphaFold2 predicted inward-open state, the authors propose a native-mechanism for Mtb EfpA being a PG-lipid floppase. Additional evidence which supports this hypothesis comes from an expanded PROSPECT analysis, which identified multiple genes associated with PG synthesis, which were sensitized upon inhibition of EfpA - a novel finding.

Though the work in this manuscript appears rigorous, there are serious concerns regarding the novelty, which I believe must first be addressed before further scientific review of the scientific work. First and foremost, the structure of apo Mtb EfpA, and BRD-8000.3 bound Mtb EfpA were already published in Nature Communications recently by Wang et al. (<https://doi.org/10.1038/s41467-024-51948-9>). Furthermore, Wang et al used molecular docking to identify binding-poses of BRD-9327, highlighting the same three residues which are identified by the cryo-EM structure from Kumar Khandelwal et al. Both Wang et al. and this manuscript identify similar, if not identical lipid binding regions, inhibitor binding sites, and propose a mechanism of lipid floppase activity. While, again, I find the work in this manuscript overall well done and rigorous, the work is simply too similar, and the lack of any reference to Wang et al. is, at the least, suspect.

I find the independent reproducibility between these two studies as a strength. Though there are aspects between this work and Wang et al. which differ (namely EfpA-BRD-9327, and identification of PG related genes), I prefer to postpone any more detailed scientific evaluation until these significant issues regarding novelty and comparison to Wang et al.'s findings are addressed.

Version 1:

Reviewer comments:

Reviewer #1

(Remarks to the Author)

This is my second review of the publication entitled: Structure and inhibition mechanisms of Mycobacterium tuberculosis essential transporter efflux protein A.

Thank you for including the lipidomics data and answering my questions. I especially appreciate the explanation about the different lipids found in different organisms in the efflux pump cavity. All my questions have been answered.

Reviewer #2

(Remarks to the Author)

The authors addressed my concerns sufficiently in the revised manuscript.

Reviewer #3

(Remarks to the Author)

Kumar Khandelwal et al have satisfactorily answered and address my concerns and thus greatly improved their manuscript, which I can now fully support !

We are grateful to the reviewers for useful suggestions. I believe that we have addressed them all as described below citing each reviewer comment and then our **Response:**

Reviewer #1 (Remarks to the Author)

The authors of this manuscript evaluated the mechanism of inhibition of two inhibitors of the EfpA efflux pump of Mtb that were recently discovered by chemical-genetic screening, thereby exploring the previously uncharacterised biomolecules that appear to be transported across the membrane by the pump. It is clear from the literature that EfpA is associated with drug resistance and is required by Mtb, making it a perfect target for future studies. This makes it an interesting topic, especially given the severity of molecular mimicry by Mtb and the lack of suitable antibiotics. This review is based on my expertise in the field of lipidomics.

In general, there is a lack of specificity in the evaluation of your lipidomics measurement. I would have expected an LC-MS measurement with subsequent data analysis of the peaks with MSMS fragmentation interpretation via software or manually. As this lipidomics measurement is part of your further explanation of how the inhibitor potentially blocks the lipid transport of the pump, the lipid data are an essential part.

Response: We agree with the reviewer that we did not provide sufficient information for our lipidomics analysis in the original submission. To address this concern, additional MS data are included in the revised manuscript, including the mass spectra of the molecular ion [M-H]⁻ species of phosphatidylglycerol (PG), cardiolipin (CL) and phosphatidyl-ethanolamine (PE) and their representative MS/MS fragmentation spectra. This is now summarized in Extended data Fig. 5. The lipid structural identification was performed manually and was based on exact mass measurement and MS/MS fragmentation.

The purpose of our lipidomic analysis is to provide qualitative assessment of which lipids may occupy the tunnels formed within the recombinant protein isolated from *E. coli*. Given this objective and other considerations (such as potentially non-specific protein-lipid binding), we didn't pursue rigorous lipid quantification, which we are keenly aware would require the use of internal standards (preferably isotope-labeled) and replication.

Major comments:

- Even if it is only in the extended data, it is not sufficient to show a chromatogram as a result. It would be better to give areas instead.

Response: We respectfully disagree that peak areas would be helpful. Due to the variations in ionization efficiencies of different lipids, and without normalization by internal lipid standards, peak areas could be very misleading.

- The method states that the lipids were measured in negative ionisation mode. Did you

also check positive ionisation, as some lipids can only be measured in one mode and others in both?

Response: The composition and relative abundance of major lipids of *E. coli* have been well characterized and reported. All major phospholipids of *E. coli* (PG, CL and PE) can be readily measured in the negative ion mode. While we agree that some lipids can be ionized more efficiently in the positive mode, any quantitative analysis by MS requires the use of respective internal standards.

- How many replicates were measured to conclude that PG is the most abundant lipid class? In another paper, currently on Researchsquare (10.21203/rs.3.rs-3740027/v1.), the authors hypothesise that there are not three PGs, but 2 PE and 1 cardiolipin binding at certain regions. As you have already discussed, 2 of the 3 PGs detected could also represent a cardiolipin. Since both your and the other manuscript state that the exact lipids responsible remain elusive, please briefly discuss the lipid combination presented in the other manuscript.

Response: Given that the purpose of our lipidomic analysis is to qualitatively assess which lipids contribute to the observed extra density, rather than precise quantification, we did not perform replication experiments.

The difference between ours and the other manuscript is that different expression systems were used; ours is *E. coli*, the other is human embryonic kidney cells, which may contribute to the differences in bound lipids. Now a third manuscript came in PNAS (October 23, 2024) where the EfpA structure is solved from EfpA expressed in yeast (*S. pombe*) (<https://doi.org/10.1073/pnas.2412653121>) where phosphatidic acid, a different lipid than ours, or that of Wang et. al; was found in the same cavity of EfpA. We expect that future study can be improved by identifying lipids found when expressed in native mycobacterial membranes.

- In the chromatogram, since you have defined the peaks as lipid classes, what specific lipid species have you detected?

Response: We have now included the mass spectra and representative MS/MS data of PG, CL and PE molecular species in the revised manuscript (Extended DATA Fig. 5).

Extended DATA Fig. 5] LC/MS analysis of lipids bound to the recombinant EfpA protein purified from *E. coli*. Negative ion mass spectra of the [M-H]⁻ ions of the molecular species of **a**, PG, **b**, PE and **c**, CL. In the parentheses, the numbers before the colon denote the carbon numbers of acyl chains, while the numbers after the colon denote the numbers of double bonds (or cyclopropanes). Representative negative ion MS/MS spectra of **d**, PG (16:0/17:1), **e**, PE (16:0/17:1) and **f**, CL (68:2). The acyl compositions are revealed by fatty acid anions shown in the product ion mass spectra.

- Was a standard included? To normalise the data it would be nice to have a standard for each lipid class included in the measurement.

Response: Given the fact that three major phospholipids (PG, CL and PE) are well characterized in *E. coli.*, and our analysis by high-resolution tandem MS was qualitative (not quantitative), we did not include lipid standards in the analysis.

Minor comments:

- Since a QTOF was used for MS detection, was it run in scan mode only to detect all lipids? A subsequent MRM acquisition would have allowed additional quantification. This could also be considered in a future study.

Response: Yes, the MS analysis by the Q-Tof instrument was acquired in a scan-mode to detect all lipids, included in the Extended DATA Fig. 5 legend.

Reviewer #2 (Remarks to the Author)

The manuscript by Khandelwal et al. reports the structures of Mycobacterium tuberculosis essential transporter efflux protein A (EfpA) with and without bound inhibitors. The structures obtained in the absence of the inhibitors reveal clear densities potentially corresponding to bound lipid molecules derived from the expression host. Based on the structural data along with the molecular dynamics simulations and lipidomic analysis, authors propose that the native substrate of EfpA could be phospholipids, specifically phosphatidylglycerol (PG). The structures in the presence of inhibitors BRD-8000.3 and BRD-9327 reveal the binding sites for these compounds through clear cryo-EM densities.

Overall, the structural data presented is strong and deserve publication in Nature Communications. While the manuscript contains substantial number of speculations that cannot readily be explained by the available data, these parts are discussed sufficiently and clear to the reader for the most part.

- The phosphate moiety of the proposed PG2 molecule would be facing to the hydrophobic core of the membrane in the absence of the dimeric arrangement that appears to be artificial. The authors should discuss this fact in the manuscript more in detail.

Response: We discussed this issue alongside the possibility raised first by others now that the antiparallel dimer may be physiological (Li et. al; PNAS October 23, 2024, doi.org/10.1073/pnas.2412653121) though we do not advocate for that. This is presented in detail in the Discussion, paragraphs 4,5,6,7.

- The maps for the ligands should be shown with the maps of the surrounding residues to be able to judge the strength of the density.

Response: We have included the density of residues in the ligand binding pocket along with ligand as shown in updated figures (Fig. 2e for BRD8000.3 and Fig. 3d for BRD9327).

- How was the alphafold models were obtained. No specifics were included in the manuscript.

Response: We obtained the AlphaFold model used as initial model for EfpA structure on cryo-EM maps from Uniprot. The Uniport website provide the link for the EfpA AlphaFold predicted model (<https://www.uniprot.org/uniprotkb/P9WJY5/entry#structure>). We have now added this information in the methods section.

- Did the authors try C2 symmetry? The resolution of the maps appears sufficient, but it may be further improved with enforcing C2 symmetry and performing symmetry expansion to perform local refinement focusing on one of the protomers within the dimer.

Response: We didn't use C2 symmetry in our data analysis. As we have a lipid molecule PG2 in both monomers whose head groups were close to each other in antiparallel dimer between the protomers (Extended DATA Fig. 4C). The center axis of C2 symmetry goes between the headgroup of PG2 molecules. Application of C2 symmetry would bring artificial signal strength at the center of axis which could show the PG2 molecules head group interacting with each other. To avoid any artificial signal amplification near the central axis we didn't apply the C2 symmetry in our data processing.

- It would be interesting to see if arginine mutated in place of P171 forms salt bridges as initially expected.

Response: Unfortunately, our data processing used a focused refinement step (EfpA transporter protein only) to improve the resolution of the transporter transmembrane helices. Hence, we could not see the BRIL-tag and Fab/nb region in the final map. However, an example of the SLC19A1 transporter structure exists where the BRIL tag was shown to make a salt bridge with an arginine of a loop of the protein (Dang et. al; Cell Discovery volume 8, Article number: 141 (2022). That we sought to mimic with our mutation in a similar position.

- Cryo-EM data processing part should be described more in detail in the methods section. As it is, no specifics were given and the readers are referred to supplementary figures.

Response: We have included further details of the data processing in the methods sections in the revised manuscript.

- Please clarify if the real space refinement was done together with the ligands. It is not clear in the methods section.

Response: The final model with ligands was obtained by real space refinement using the reference model and Ramachandran restraints, with each inhibitor present. We included this part in the methods section in the revised manuscript.

Reviewer #3 (Remarks to the Author):

In this manuscript, Kumar Khandelwal et al. resolved the structures of Mtb EfpA, a pharmacologically relevant multi-drug efflux pump, bound to the recently identified inhibitors BRD-8000.3 and BRD-9327 using single particle cryoEM. Identification of the BRD compounds was performed with the authors in-house developed PROSPECT pipeline (<https://doi.org/10.1021/acsinfecdis.9b00373>). Identification of the apo, and multiple holo conformations, i.e., BRD-8000.3, BRD-9327, and a combination of the two, provides insights as to the mechanism of the two non-competitive inhibitors of Mtb EfpA. Furthermore, with the addition of an AlphaFold2 predicted inward-open state, the authors propose a native-mechanism for Mtb EfpA being a PG-lipid floppase. Additional evidence which supports this hypothesis comes from an expanded PROSPECT analysis, which identified multiple genes associated with PG synthesis, which were sensitized upon inhibition of EfpA - a novel finding.

-Though the work in this manuscript appears rigorous, there are serious concerns regarding the novelty, which I believe must first be addressed before further scientific review of the scientific work. First and foremost, the structure of apo Mtb EfpA, and BRD-8000.3 bound Mtb EfpA were already published in Nature Communications recently by Wang et al. (<https://doi.org/10.1038/s41467-024-51948-9>).

Response: We now discuss their results at length in the Discussion section.

-Furthermore, Wang et al used molecular docking to identify binding-poses of BRD-9327, highlighting the same three residues which are identified by the cryo-EM structure from Kumar Khandelwal et al. Both Wang et al. and this manuscript identify similar, if not identical lipid binding regions, inhibitor binding sites, and propose a mechanism of lipid floppase activity.

Response: We concur with Wang et al in respect of lipid binding sites. Lipid identification is not definitive in any of the cases yet, relying on lipids from different expression cells, and mass spectrometry that determines the lipids and chain lengths present in the samples, but cannot quantitate which ones dominate stoichiometrically. As to BRD 9327, The three residues against which Wang et al docking was performed were identified earlier by mutagenesis versus drug efficacy of BRD9327 from the coauthors in our manuscript who discovered the compounds in 2020 (Deb Hung group) (Johnson et. al; ACS Infect. Dis. 2020, 6, 56-63). Wang et al focused their computer docking specifically against just those residues. However, their focused computational docking turned out to be incorrect.

Our determined structure is independent of any prior knowledge of where the site might be, and the orientation of BRD 9327 is 90°, ie completely different from their dock solution. We now make this clear in the manuscript.

-While, again, I find the work in this manuscript overall well done and rigorous, the work is simply too similar, and the lack of any reference to Wang et al. is, at the least, suspect.

Response: Our work was completed before the appearance of the Res Sq version of Wang et al. 2024 that has now been published in a refereed journal (Nature communications 2024). As the reviews of our submission took time.

-I find the independent reproducibility between these two studies as a strength. Though there are aspects between this work and Wang et al. which differ (namely EfpA-BRD-9327, and identification of PG related genes), I prefer to postpone any more detailed scientific evaluation until these significant issues regarding novelty and comparison to Wang et al.'s findings are addressed.

Response: We have now addressed the findings of Wang et. al; and those of a very recent paper in PNAS on the EfpA structure by Li et al. 2024. In short, we all have similar structures from expression in different organisms, thus the bound lipids may be different derived from the different expression systems. The notions of whether or not the asymmetric dimer of EfpA is functional we address in the Discussion section paragraphs 4-7. Our manuscript is the only manuscript that show the structure determination of EfpA inhibition by BRD-9327, and in combination together with BRD 8000.3, that in turn validates the combination therapy approach.

Details for the record:

The paper by Wang et al was published on Jan 5th on Nature Research Square (unrefereed) , then September 4th in Nature Comms. There is also now a third publication from different authors that was published on October 23rd, 2024, from Li et al in PNAS. Our manuscript was submitted to Nature on June 11th (tracking number: 2024-06-11950) transferred to Nature Structure and Molecular Biology on July 17th, (Receipt of NSMB-A49550) and transferred to Nature Communications on August 9-13. We received the first two reviews on October 7th, and the third review on November 11th. Hence our work has been over 3 months at Nature Comms, following 2 months at Nature journals prior to that. We have now incorporated citation of all relevant publications up to this date of Nov 20th, 2024.